# Coordinated cadherin functions sculpt respiratory motor circuit connectivity

Alicia N Vagnozzi[1], Matthew T Moore[1], Minshan Lin[1], Elyse M Brozost[1], Ritesh KC[1], Aambar Agarwal[1], Lindsay A Schwarz[2], Xin Duan[3], Niccolò Zampieri[4], Lynn T Landmesser[1], Polyxeni Philippidou[1]*

[1]Department of Neurosciences, Case Western Reserve University, Cleveland, United States; [2]Department of Developmental Neurobiology, St. Jude Children's Research Hospital, Memphis, United States; [3]Department of Ophthalmology, University of California, San Francisco, San Francisco, United States; [4]Max Delbrück Center for Molecular Medicine in the Helmholtz Association, Berlin, Germany

**Abstract** Breathing, and the motor circuits that control it, is essential for life. At the core of respiratory circuits are Dbx1-derived interneurons, which generate the rhythm and pattern of breathing, and phrenic motor neurons (MNs), which provide the final motor output that drives diaphragm muscle contractions during inspiration. Despite their critical function, the principles that dictate how respiratory circuits assemble are unknown. Here, we show that coordinated activity of a type I cadherin (N-cadherin) and type II cadherins (Cadherin-6, -9, and -10) is required in both MNs and Dbx1-derived neurons to generate robust respiratory motor output. Both MN- and Dbx1-specific cadherin inactivation in mice during a critical developmental window results in perinatal lethality due to respiratory failure and a striking reduction in phrenic MN bursting activity. This combinatorial cadherin code is required to establish phrenic MN cell body and dendritic topography; surprisingly, however, cell body position appears to be dispensable for the targeting of phrenic MNs by descending respiratory inputs. Our findings demonstrate that type I and II cadherins function cooperatively throughout the respiratory circuit to generate a robust breathing output and reveal novel strategies that drive the assembly of motor circuits.

*For correspondence:
pxp282@case.edu

Competing interest: The authors declare that no competing interests exist.

## Editor's evaluation

The overarching hypothesis is that cadherin adhesion molecules specify the code that enables the premotor brainstem breathing circuits to innervate the phrenic motor neurons that control the primary breathing muscle, the diaphragm. This concept is important for understanding how the breathing control circuit is established and in general, how motor circuitry is developed. This is an extremely thorough investigation of the role of cadherins in generating a functional motor circuit.

## Introduction

The ability to breathe is necessary for life. Breathing is generated by complex neural networks and cellular mechanisms in the brainstem and spinal cord that ultimately control muscle contraction to produce respiratory-related movement. Respiratory rhythmogenesis is initiated by an oscillatory population in the brainstem, the preBötzinger (preBötC) complex (*Del Negro et al., 2018*). This respiratory rhythm is relayed through an area in the brainstem called the rostral Ventral Respiratory Group (rVRG) to Phrenic Motor Column (PMC) neurons in the spinal cord (*Wu et al., 2017*). PMC neurons provide the sole innervation to the diaphragm, a muscle that is critical for bringing oxygenated air into the lungs during inspiration (*Greer, 2012*). While the ability of phrenic motor neurons (MNs) to receive

**eLife digest** The neural circuits which control breathing are established in the womb, ready to switch on with the first gulp of air. Defects in the way that this network is assembled can result in conditions such as sudden infant death syndrome. This process, however, remains poorly understood; in particular, it is still unclear exactly how the two main types of nerve cells which form respiratory circuits start to 'talk' to each other.

Known as Dbx1-derived interneurons and phrenic motor neurons, these cell populations reside in different parts of the body and perform distinct roles. The interneurons, which are present in the brainstem, act as a pacemaker to set the rhythm of respiration; the motor neurons reside in the spinal cord, connecting the interneurons with the muscles which allow the lungs to fill with air.

Vagnozzi et al. aimed to identify how phrenic motor neurons connect to and relay signals from other neurons involved in breathing to the diaphragm muscle. To do so, the team focused on cadherins, a group of proteins which allow cells to attach to one another. Studded through the membrane, these molecules are also often involved in forming connections from one cell to another that allow them to communicate.

Newborn mice in which phrenic motor neurons lacked a specific combination of cadherins experienced respiratory failure, showing that these proteins were needed for breathing circuits to develop normally. Electrical activity recorded from these cells showed that phrenic motor neurons lacking cadherins could not receive the signals required to activate the breathing muscles. Microscopy imaging also revealed that the loss of cadherins shifted the position of the phrenic motor neurons within the spinal cord; however, this change did not seem to affect the connections these cells could establish.

The ability to breathe is compromised in many incurable human diseases such as muscular dystrophies and amyotrophic lateral sclerosis. It may be possible to alleviate some of these symptoms by integrating phrenic motor neurons created in the laboratory into existing circuits. Studies which aim to decipher how the respiratory network is established, such as the one conducted by Vagnozzi et al., are essential in this effort.

and integrate descending inputs from the rVRG is essential for breathing, the molecular mechanisms that underlie rVRG–PMC connectivity are largely unknown.

PMC neurons are a specialized subset of MNs confined to the cervical region of the spinal cord. Unlike the majority of other MN subtypes, they largely eschew propriospinal inputs and instead integrate medullary inputs to produce robust respiratory activity (*Wu et al., 2017*). The transcriptional programs that define molecular features of phrenic MNs which distinguish them from other MN populations during development are beginning to emerge (*Chaimowicz et al., 2019*; *Machado et al., 2014*; *Philippidou et al., 2012*; *Vagnozzi et al., 2020*). Phrenic-specific transcription factors (TFs) deploy molecular programs that establish their unique morphology and stereotyped position, but whether these features of their identity contribute to their selective connectivity with excitatory premotor respiratory populations is not well understood. In the monosynaptic stretch reflex circuit, where group Ia sensory afferents synapse directly onto the alpha MNs innervating the same muscle with exquisite specificity, both MN cell body position and dendritic orientation have been implicated in controlling connectivity patterns during development (*Balaskas et al., 2019*; *Sürmeli et al., 2011*). While the stereotyped phrenic MN position likely contributes to circuit formation, the non-laminar architecture of the spinal cord might render a topography-based targeting strategy insufficient for precise connectivity. While the long-standing chemoaffinity hypothesis, proposing that cell surface homophilic interactions enable matching between synaptic partners (*Sperry, 1963*), has been illustrated in the visual and olfactory circuits (*Graham and Duan, 2021*; *Xie et al., 2022*), the relative contribution of transmembrane recognition molecules in motor circuits, and to phrenic MN connectivity specifically, is unclear.

Phrenic MNs express a unique combination of cell surface adhesion molecules that could potentially serve as molecular recognition tags for descending brainstem axons (*Machado et al., 2014*; *Vagnozzi et al., 2020*). We previously identified a distinct combinatorial cadherin code that defines phrenic MNs, which includes both the broadly expressed type I N-cadherin and a subset of specific type II

cadherins (*Vagnozzi et al., 2020*). Cadherins establish the segregation and settling position of MN cell bodies in the spinal cord (*Demireva et al., 2011*; *Dewitz et al., 2019*; *Dewitz et al., 2018*; *Price et al., 2002*), but their contribution to the wiring of motor circuits and their function has not yet been determined. In the retina, type II cadherins direct neuronal projections to distinct laminae, thereby dictating synaptic specificity (*Duan et al., 2014*; *Duan et al., 2018*; *Osterhout et al., 2011*), while in the hippocampus they establish synaptic fidelity without overtly affecting neuronal morphology (*Basu et al., 2017*). Despite utilizing distinct strategies to establish connectivity, in both the retina and the hippocampus type II cadherins function independently, without any contributions from type I family members. In contrast, both type I and II cadherins are required for establishing the stereotyped motor pool organization in the spinal cord. This raises the question of whether type I and II cadherins may have other unique coordinated functions affecting the development and function of motor circuits.

Here, we show that coordinated type I and II cadherin signaling is necessary for robust respiratory output. After MN-specific deletion of cadherins N, 6, 9, and 10, mice display severe respiratory insufficiency, gasp for breath, and die within hours of birth. Using phrenic nerve recordings, we determined that cadherins are crucial for respiratory motor output, as MN-specific cadherin inactivation leads to a striking decrease in phrenic MN activity. We further show that cadherins establish phrenic MN cell body positioning and dendritic orientation, but surprisingly find that cell body positioning is likely dispensable for proper phrenic MN firing. Finally, we show that cadherin signaling is required in Dbx1-derived interneurons, which give rise to the premotor rVRG, for phrenic MN bursting activity. Collectively, our results demonstrate that cadherins are vital to respiratory circuit assembly and function, and suggest a model where cadherin-mediated adhesive recognition dictates wiring specificity through multifaceted functions in dendritic organization and molecular recognition.

## Results

### A combinatorial cadherin code establishes breathing and is required for life

Phrenic MNs express a distinct combinatorial code of cell surface transmembrane molecules (*Machado et al., 2014*; *Philippidou et al., 2012*; *Vagnozzi et al., 2020*), including a subset of cadherins, but the function of these molecules in phrenic MN development, connectivity, and function has not been established. We previously conducted in situ hybridization for all type I and II cadherins in the PMC at e13.5, a timepoint at which early phrenic MN topography is already established, but synapse formation has yet to occur. We found that phrenic MNs express the type I *N-cadherin* (*Cdh2*) and the type II cadherins *Cdh6*, *9*, *10*, *11*, and *22* (*Vagnozzi et al., 2020*). While the type I cadherin *N-cadherin* and the type II cadherins *Cdh6*, *11*, and *22* show broad expression in either all neurons (*N-cadherin*) or all MNs (*6*, *11*, and *22*), *Cdh9* and *10* are remarkably specific to phrenic MNs (*Figure 1a*, *Figure 1—figure supplement 1a*). We also found that loss of cadherin signaling specifically in MNs, via inactivation of the cadherin obligate intracellular partners β- and γ-catenin, leads to embryonic lethality, suggesting a critical role for cadherins in phrenic MN function (*Vagnozzi et al., 2020*).

Since cadherins 9 and 10 show PMC-specific expression, we thought they were likely to contribute to respiratory circuit assembly. While cadherins are known to engage homophilic interactions, recent evidence has emerged also demonstrating strong heterophilic adhesive recognition between defined subsets of type II cadherins. PMC-expressed cadherins 6, 9 and 10 form a specificity group, preferentially interacting with each other compared to other cadherins (*Brasch et al., 2018*). Therefore, we analyzed $Cdh6-/-;Cdh9-/-;Cdh10-/-$ mice (referred to as $6910^{KO}$ mice) to define the contributions of these cadherins to phrenic MN development (*Duan et al., 2018*). Surprisingly, we found that neonatal (P2) $6910^{KO}$ mice have normal minute ventilation (volume of air inhaled per minute) and live to adulthood (*Figure 1—figure supplement 2b*).

It has recently been demonstrated that type I cadherins, such as N-cadherin, can mask the contributions of type II cadherins in controlling MN cell body positioning, raising the possibility that type I and II cadherins may have coordinated functions in other aspects of MN development (*Dewitz et al., 2019*). We therefore obtained conditional $Cdh2^{flox/flox}$ mice (*Kostetskii et al., 2005*) and crossed them to $Olig2^{Cre}$ mice to inactivate N-cadherin specifically in MNs ($Cdh2^{flox/flox};Olig2^{Cre}$, referred to as $N^{MN\Delta}$ mice). We then bred $N^{MN\Delta}$ and $6910^{KO}$ mice to eliminate four out of the six cadherins expressed in phrenic MNs ($Cdh2^{flox/flox};Cdh6-/-;Cdh9-/-;Cdh10-/-;Olig2^{Cre}$, referred to as $N^{MN\Delta}6910^{KO}$ mice), and

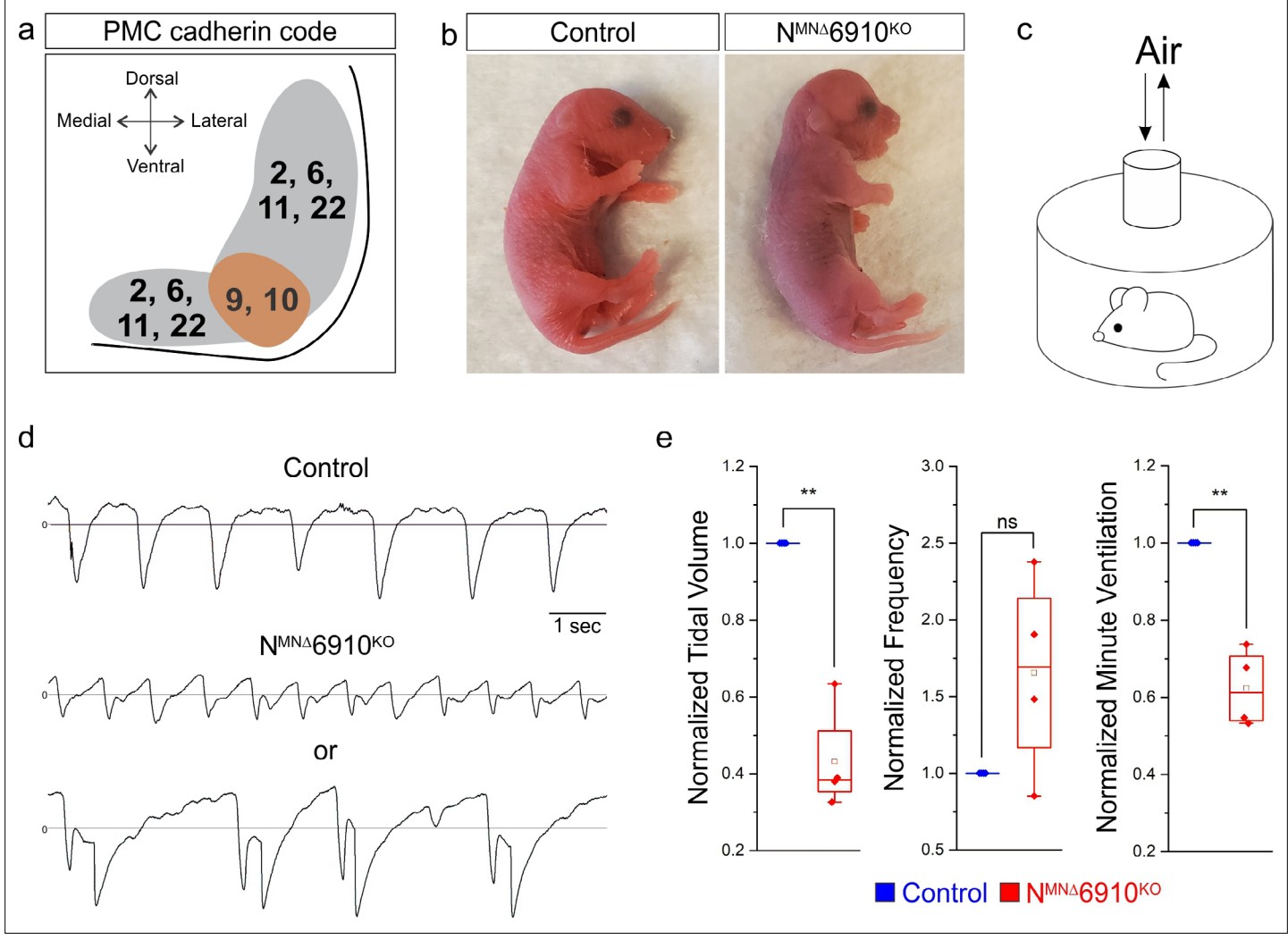

**Figure 1.** A combinatorial cadherin code establishes breathing and is required for life. (**a**) A combinatorial cadherin code defines phrenic MNs during development. Reproduced from Figure 4c from **Vagnozzi et al., 2020**. The ventral lateral spinal cord at the level of the PMC (brown circle) is depicted (see also **Figure 1—figure supplement 1a**). (**b**) To determine the function of cadherins in respiratory motor circuits, we specifically inactivated a combination of type I and II cadherins in MNs- N-cadherin and Cadherins 6, 9, 10 ($Cdh2^{flox/flox};Olig2^{Cre};Cdh6-/-;Cdh9-/-;Cdh10-/-$, referred to as $N^{MN\Delta}6910^{KO}$ mice). $N^{MN\Delta}6910^{KO}$ mice are cyanotic, appear to gasp for breath, and die shortly after birth. (**c**) Experimental setup for whole body plethysmography experiments (see also **Figure 1—figure supplement 2a**). (**d**) Representative 10s traces in room air from control and $N^{MN\Delta}6910^{KO}$ mice at P0. $N^{MN\Delta}6910^{KO}$ mice either exhibit fast, shallow breathing (middle), or irregular gasping behavior (bottom). (**e**) $N^{MN\Delta}6910^{KO}$ mice display reduced tidal volume and increased variability in respiratory frequency, resulting in a 40% reduction in overall ventilation ($n = 4$ for each genotype). **$p < 0.01$, unpaired, two-tailed Student's $t$-test.

The online version of this article includes the following video, source data, and figure supplement(s) for figure 1:

**Source data 1.**

**Figure supplement 1.** Validation of $N^{MN\Delta}6910^{KO}$ mice.

**Figure supplement 2.** Normal breathing behaviors in $6910^{KO}$ mice.

**Figure supplement 3.** Cadherins N, 6, 9, and 10 control diaphragm innervation.

**Figure 1—video 1.** Respiratory insufficiency in $N^{MN\Delta}6910^{KO}$ mice.

https://elifesciences.org/articles/82116/figures#fig1video1

**Figure 1—video 2.** Functional neuromuscular junctions (NMJs) in $N^{MN\Delta}6910^{KO}$ mice.

https://elifesciences.org/articles/82116/figures#fig1video2

to circumvent compensation of Cdh6/9/10 function by N-cadherin. $N^{MN\Delta}6910^{KO}$ mice lack *N-cadherin* and *Cdh6*, *9*, and *10* expression in all MNs by e11.5 (*Figure 1—figure supplement 1b*).

We found that both $N^{MN\Delta}$ and $N^{MN\Delta}6910^{KO}$ mice die within 24 hr of birth. While $N^{MN\Delta}$ mice likely die due to deficiencies in feeding (data not shown), $N^{MN\Delta}6910^{KO}$ mice appear cyanotic and gasp for breath (*Figure 1b*, *Figure 1—video 1*). In order to assess breathing in $N^{MN\Delta}6910^{KO}$ mice, we utilized unrestrained whole body plethysmography (*Figure 1—figure supplement 2b*, *Figure 1c*). We found that $N^{MN\Delta}6910^{KO}$ mice displayed two different kinds of abnormal respiratory behavior: either fast and shallow breathing, or slow, irregular, gasping behavior (*Figure 1d*). During the periods of shallower breathing, tidal volume (the amount of air inhaled during a single breath) was reduced by over 50%, while the frequency of breathing was variable but not statistically different from control (*Figure 1e*). Together, these two parameters combined resulted in a 40% reduction in minute ventilation (volume of air inhaled per minute, *Figure 1e*), indicating that $N^{MN\Delta}6910^{KO}$ mice have severe respiratory insufficiency and die due to respiratory failure. Our findings indicate that the combinatorial activity of N-cadherin, Cdh6, 9, and 10 is required for proper respiratory behavior.

We first asked whether diaphragm innervation defects may explain the reduction in tidal volume in $N^{MN\Delta}6910^{KO}$ mice. We examined diaphragm innervation in control, $6910^{KO}$, $N^{MN\Delta}$, and $N^{MN\Delta}6910^{KO}$ mice at e18.5. We found that $6910^{KO}$ mice have normal diaphragm innervation, while $N^{MN\Delta}$ mice have a slight decrease (~16%) in terminal arborization in the ventral diaphragm (*Figure 1—figure supplement 3a–c*). $N^{MN\Delta}6910^{KO}$ mice show a significant reduction in diaphragm innervation (~30% loss of innervation), which is more pronounced at the ventral part of the diaphragm (*Figure 1—figure supplement 3a–c*, arrows). We next asked whether the neuromuscular junctions (NMJs) that were present at the diaphragm were functional and able to generate muscle contraction. We stimulated the phrenic nerve in reduced medullary-brainstem preparations at e18.5/P0, and observed similar contractions in control and $N^{MN\Delta}6910^{KO}$ mice, indicating functional NMJs and suggesting that the phrenic nerve retains its ability to contract the diaphragm in $N^{MN\Delta}6910^{KO}$ mice (*Figure 1—video 2*). While diaphragm innervation defects may partially contribute to the tidal volume reduction and perinatal lethality in $N^{MN\Delta}6910^{KO}$ mice, we have previously shown that as little as 40% diaphragm innervation is sufficient for survival (*Vagnozzi et al., 2020*). Therefore, it is unlikely that ~30% reduction in diaphragm innervation solely underlies the respiratory insufficiency seen upon cadherin inactivation.

## N-cadherin predominantly establishes phrenic MN cell body position

What accounts for the perinatal lethality in $N^{MN\Delta}6910^{KO}$ mice? We first asked whether early phrenic MN specification, migration, and survival are impacted after cadherin inactivation. We acquired transverse spinal cord sections through the entire PMC at e13.5 and stained for the phrenic-specific TF Scip and the MN-specific TF Isl1/2, to label all phrenic MNs. We found a cluster of Scip + MNs in the ventral cervical spinal cord of control, $6910^{KO}$, $N^{MN\Delta}$, and $N^{MN\Delta}6910^{KO}$ mice, indicating that early phrenic MN specification is unperturbed (*Figure 2—figure supplement 1a*). We noted that both $N^{MN\Delta}$ and $N^{MN\Delta}6910^{KO}$ mice occasionally show migratory defects, where a few phrenic MNs remain close to the midline instead of fully migrating (*Figure 2—figure supplement 1b*, arrow). In addition, $N^{MN\Delta}$ and $N^{MN\Delta}6910^{KO}$ mice showed a similar decrease in phrenic MN numbers, likely from the loss of trophic support due to the decrease in diaphragm innervation (*Figure 2—figure supplement 1c*). While phrenic MN loss may contribute to the perinatal lethality observed in $N^{MN\Delta}6910^{KO}$ mice, we have previously found that, even after substantial MN loss, 50% of surviving phrenic MNs are sufficient to support life (*Vagnozzi et al., 2020*).

While phrenic MNs are normally distributed along the rostrocaudal axis, we noticed that they appear to shift ventrally in $N^{MN\Delta}$ and $N^{MN\Delta}6910^{KO}$ mice. To quantitate PMC cell body position, each phrenic MN was assigned a Cartesian coordinate, with the midpoint of the spinal cord midline defined as (0,0). We first examined control and $6910^{KO}$ mice and found no difference in PMC position (*Figure 2a, b*, *Figure 2—figure supplement 1a*). Both $N^{MN\Delta}$ and $N^{MN\Delta}6910^{KO}$ mice displayed a significant shift in phrenic MN cell body position, with cell bodies shifting ventrally toward the edge of the spinal cord (*Figure 2c–f*, *Figure 2—figure supplement 1a*). Cell bodies in $N^{MN\Delta}$ and $N^{MN\Delta}6910^{KO}$ mice overlapped when compared on contour density plots (*Figure 2g, h*). These findings were further supported after quantification of the average ventrodorsal and mediolateral phrenic MN position per embryo (*Figure 2i–l*). Correlation analysis determined that control and $6910^{KO}$ mice are highly similar ($r = 0.95$), as are $N^{MN\Delta}$ and $N^{MN\Delta}6910^{KO}$ mice ($r = 0.96$), but each group is dissimilar from the

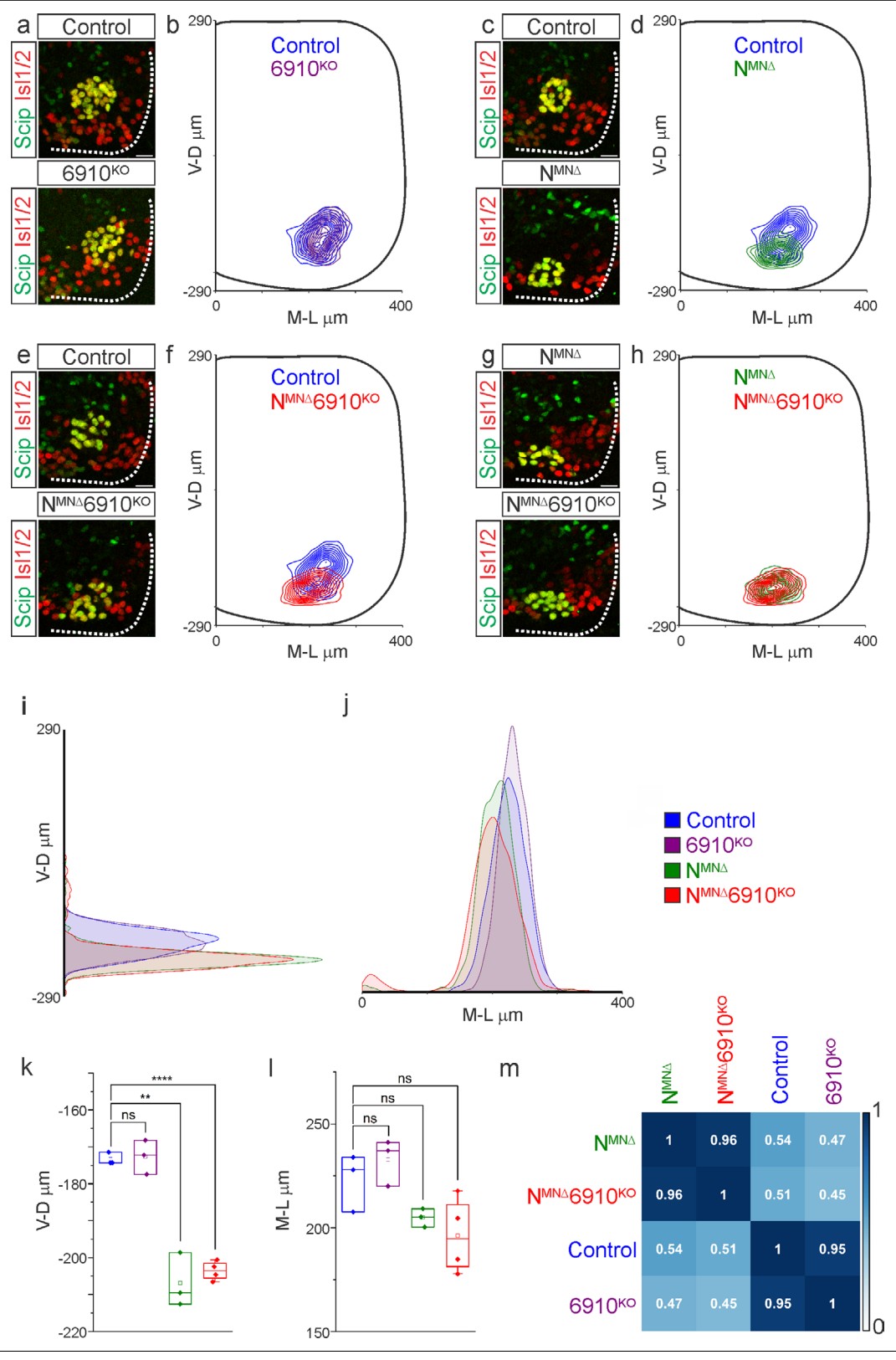

**Figure 2.** N-cadherin predominantly establishes phrenic motor neuron (MN) cell body position. Analysis of cell body position reveals differential contributions of type I and II cadherins to phrenic motor topography. (**a, c, e, g**) Phrenic MN cell bodies (yellow, defined by the co-expression of the phrenic-specific TF Scip in green and the MN-specific TF Isl1/2 in red) are located at the same position in control and 6910[KO] mice, but show a similar

*Figure 2 continued*

ventral shift in both N$^{MNΔ}$ and N$^{MNΔ}$6910$^{KO}$ mice at e13.5. Scale bar = 25 μm. (**b, d, f, h**) Contour density plot of cell body position in control, 6910$^{KO}$, N$^{MNΔ}$, and N$^{MNΔ}$6910$^{KO}$ mice at e13.5. N$^{MNΔ}$ and N$^{MNΔ}$6910$^{KO}$ mice display a similar ventral shift in phrenic MN position, suggesting that additional deletion of type II cadherins does not exacerbate positional changes caused by loss of N-cadherin. V-D μm: ventrodorsal position, M-L μm: mediolateral position. (0,0) represents the center of the spinal cord in both dimensions. Density plots of ventrodorsal (**i**) and mediolateral (**j**) cell body position in control, 6910$^{KO}$, N$^{MNΔ}$, and N$^{MNΔ}$6910$^{KO}$ mice. Quantification of ventrodorsal (**k**) and mediolateral (**l**) position in control, 6910$^{KO}$, N$^{MNΔ}$ and N$^{MNΔ}$6910$^{KO}$ mice. Cell bodies in N$^{MNΔ}$ and N$^{MNΔ}$6910$^{KO}$ mice display a statistically significant ventral shift. (**m**) Correlation analysis of phrenic MN positional coordinates in control, 6910$^{KO}$, N$^{MNΔ}$, and N$^{MNΔ}$6910$^{KO}$ mice. 0 is no correlation, while 1 is a perfect correlation (*n* = 1442 control, *n* = 1422 6910$^{KO}$, *n* = 1052 N$^{MNΔ}$, and *n* = 1220 N$^{MNΔ}$6910$^{KO}$ somas from *n* = 3 control, *n* = 3 6910$^{KO}$, *n* = 3 N$^{MNΔ}$, and *n* = 4 N$^{MNΔ}$6910$^{KO}$ mice). **p < 0.01, ****p < 0.0001, unpaired, two-tailed Student's *t*-test.

The online version of this article includes the following source data and figure supplement(s) for figure 2:

**Source data 1.**

**Figure supplement 1.** N-cadherin predominantly establishes phrenic motor neuron (MN) cell body position.

other (*r* = 0.45–0.54, *Figure 2m*). Therefore, 6910$^{KO}$ mice show no positional changes, while N$^{MNΔ}$ and N$^{MNΔ}$6910$^{KO}$ mice display similar changes in cell body position, suggesting a predominant function for N-cadherin in phrenic MN topography.

## Phrenic MN dendritic orientation requires cadherins N, 6, 9, and 10

Surprisingly, our cell position analysis indicated that type II cadherins are dispensable for phrenic MN position. Yet, type II cadherin expression uniquely defines phrenic MNs and only N$^{MNΔ}$6910$^{KO}$ mice appear to die due to respiratory insufficiency. We therefore asked whether any other properties of phrenic MNs might rely on cooperative actions of N-cadherin and type II cadherins. We examined dendritic orientation in control, 6910$^{KO}$, N$^{MNΔ}$, and N$^{MNΔ}$6910$^{KO}$ mice by injecting the lipophilic dye DiI into the phrenic nerve. DiI diffuses along the phrenic nerve to label both PMC cell bodies and dendrites. In control and 6910$^{KO}$ mice, phrenic MN dendrites branch out in dorsolateral to ventrome-dial directions (*Figure 3a, c*, *Figure 3—figure supplement 1a, b*). In N$^{MNΔ}$ mice, dendrites appear ventralized and do not reach as far in the dorsolateral direction (*Figure 3e*, *Figure 3—figure supplement 1c*). Phrenic MN dendrites in N$^{MNΔ}$6910$^{KO}$ mice show a dramatic reorganization, exhibiting both ventralization and defasciculation (*Figure 3g*, *Figure 3—figure supplement 1d, e*).

To quantify these changes, we superimposed a radial grid divided into octants onto the dendrites and measured the fluorescence intensity in each octant after removal of any fluorescence associated with the cell bodies. Zero degrees was defined by a line running perpendicular from the midline through the center of cell bodies. In control and 6910$^{KO}$ mice, the main contributions to dendritic fluorescence intensity came from dorsolaterally projecting dendrites (0–90 degrees), representing 40–45% of the overall dendritic intensity (*Figure 3b, d, i, m*). Ventrally projecting dendrites (180–225 and 315–360 degrees) were the next main contributors, representing nearly 30% of the overall dendritic intensity (*Figure 3b, d, i, m*). In N$^{MNΔ}$ mice, the intensity of dorsolateral dendrites was maintained (*Figure 3f, j, m*), while there was an increase in ventral dendrites (*Figure 3m*). N$^{MNΔ}$6910$^{KO}$ mice had a striking reduction in dorsolateral dendrites, with a concomitant increase in ventral dendrites (*Figure 3h, k, m*). When directly comparing N$^{MNΔ}$ and N$^{MNΔ}$6910$^{KO}$ mice, N$^{MNΔ}$6910$^{KO}$ mice had a more severe loss of dorsolateral dendrites while both N$^{MNΔ}$ and N$^{MNΔ}$6910$^{KO}$ mice showed an increase in ventral dendrites, indicating that N-cadherin predominantly acts to restrict the number of ventrally projecting dendrites (*Figure 3l, m*). Together, these data suggest that cadherins N, 6, 9, and 10 collectively control phrenic MN dendritic orientation and topography, which may contribute to their targeting by presynaptic partners.

## Coordinated type I and II cadherin signaling drives phrenic MN activation

What are the consequences of altered cell body and dendritic topography in N$^{MNΔ}$ and N$^{MNΔ}$6910$^{KO}$ mice? Both MN cell body position and dendritic orientation have been suggested to underlie MN connectivity in the spinal cord. If phrenic MNs fail to properly integrate into respiratory circuits in the absence of cadherin function, we would expect to see changes in their activity. To examine changes

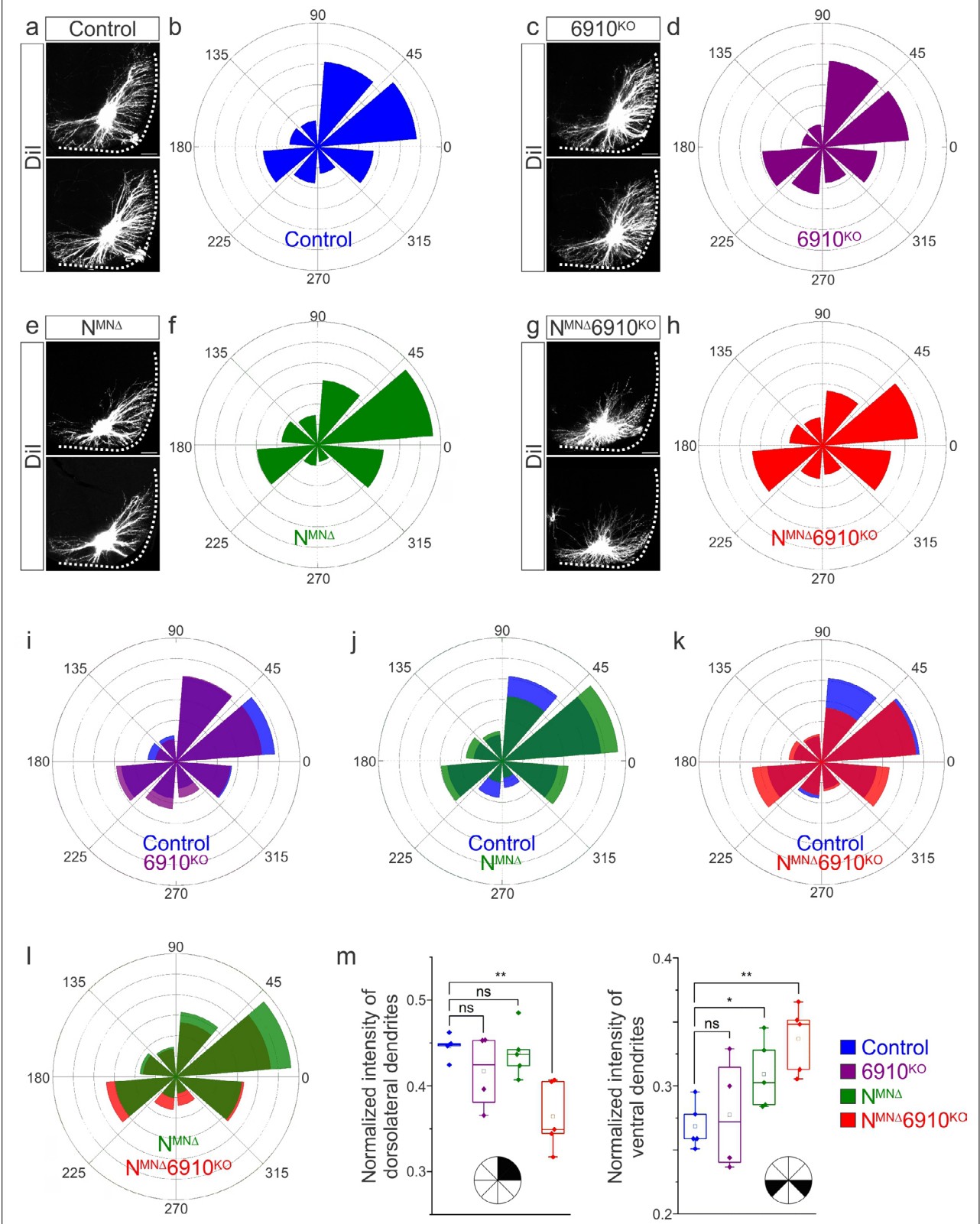

**Figure 3.** Cadherins N, 6, 9, and 10 dictate phrenic motor neuron (MN) dendritic orientation. (**a, c, e, g**) DiI injections into the phrenic nerve in control, 6910<sup>KO</sup>, N<sup>MNΔ</sup>, and N<sup>MNΔ</sup>6910<sup>KO</sup> mice reveal phrenic MN dendrites, which extend in the dorsolateral and ventromedial directions in control mice. Phrenic MN dendrites do not change in 6910<sup>KO</sup> mice, but appear to not reach as far in the dorsolateral direction and to increase ventrally in N<sup>MNΔ</sup> mice. Strikingly, in N<sup>MNΔ</sup>6910<sup>KO</sup> mice, phrenic MN dendrites appear defasciculated, have reduced dorsolateral projections, and an increase in ventral projections. Scale

*Figure 3 continued on next page*

*Figure 3 continued*

bar = 100 µm. (**b, d, f, h**) Radial plot of the normalized fluorescent intensity in each octant in control, 6910$^{KO}$, N$^{MN\Delta}$, and N$^{MN\Delta}$6910$^{KO}$ mice. Zero degrees represents a line through the center of the phrenic MN cell bodies that is perpendicular to the midline. Radial plot of the normalized fluorescent intensity in (**i**) control and 6910$^{KO}$ mice, (**j**) control and N$^{MN\Delta}$ mice, (**k**) control and N$^{MN\Delta}$6910$^{KO}$ mice, and (**l**) N$^{MN\Delta}$ and N$^{MN\Delta}$6910$^{KO}$ mice. (**m**) Quantification of the proportion of dendritic fluorescent intensity from 0 to 90 degrees (dorsolateral) and from 180 to 225 and 315 to 360 degrees (ventral, $n$ = 5 control, $n$ = 4 6910$^{KO}$, $n$ = 5 N$^{MN\Delta}$, and $n$ = 5 N$^{MN\Delta}$6910$^{KO}$ mice). *p < 0.05, **p < 0.01, unpaired, two-tailed Student's $t$-test.

The online version of this article includes the following source data and figure supplement(s) for figure 3:

**Source data 1.**

**Figure supplement 1.** Phrenic motor neuron (MN) dendritic orientation requires cadherins N, 6, 9, and 10.

to respiratory circuitry intrinsic to the brainstem and spinal cord independent of sensory input, we performed suction recordings of the phrenic nerve in isolated brainstem–spinal cord preparations (*Figure 4a*). These preparations display fictive inspiration after the removal of inhibitory networks in the pons via transection, allowing us to interrogate circuit level changes. We tested whether cadherin deletion impacts circuit output at e18.5/P0, shortly before N$^{MN\Delta}$ and N$^{MN\Delta}$6910$^{KO}$ mice die. We first examined respiratory burst frequency and duration. Burst frequency was highly variable and irregular even in control mice at e18.5/P0, as the respiratory rhythm has not yet stabilized (*Hilaire and Duron, 1999*), precluding meaningful analysis of respiratory burst frequency amongst different cadherin mutants. Burst duration, however, was more consistent, and was similar in control, N$^{MN\Delta}$, 6910$^{KO}$, and N$^{MN\Delta}$6910$^{KO}$ mice (*Figure 4b*).

In contrast, we observed a striking difference in the activation of phrenic MNs in N$^{MN\Delta}$6910$^{KO}$ mice (*Figure 4c, d*). While bursts in control, 6910$^{KO}$, and N$^{MN\Delta}$ mice exhibit large peak amplitude, bursts in N$^{MN\Delta}$6910$^{KO}$ mice were either of very low amplitude (~70%) or non-detectable (~30%). After rectifying and integrating the traces, we found a nearly 70% decrease in both total burst activity and burst activity normalized over time in N$^{MN\Delta}$6910$^{KO}$ mice (*Figure 4e, f*). Notably, N$^{MN\Delta}$ mice show normal burst activity, supporting our initial observations that their perinatal death was not due to respiratory failure. Furthermore, the fact that phrenic MNs maintain their normal activity pattern in N$^{MN\Delta}$ mice suggests that neither cell body position nor phrenic MN numbers significantly contribute to phrenic MN output. Our data also indicate that individual type I and II cadherins are dispensable for normal phrenic MN activity, but that coordinated activity of both types is imperative for robust activation of phrenic MNs during inspiration.

We next asked whether phrenic MNs in N$^{MN\Delta}$6910$^{KO}$ mice specifically lose respiratory-related activity, or if instead they lose the capacity to respond to all inputs indiscriminately. While not normally detectable, latent non-respiratory propriospinal networks can activate phrenic MNs in a pattern that is distinct from respiratory bursts under disinhibitory conditions (*Cregg et al., 2017*). This spinal cord-initiated phrenic MN activity is of lower amplitude and longer duration than respiratory bursts, and persists after C1 transection eliminates descending inputs from brainstem respiratory centers. When we bath applied the GABA$_A$ antagonist picrotoxin and the glycine antagonist strychnine in control mice, we observed spinal network activity alongside normal respiratory bursts (*Figure 4g*, respiratory bursts marked with magenta arrows). Spinal network activity after the addition of picrotoxin and strychnine was similar in control and N$^{MN\Delta}$6910$^{KO}$ mice, both in duration and normalized integrated activity over time, indicating that phrenic MNs are being activated normally by these propriospinal inputs after cadherin loss (*Figure 4h*). However, respiratory bursts remained of reduced amplitude (*Figure 4g*). Our data indicate that phrenic MNs are normally integrated into propriospinal circuits and, despite the reduction in respiratory activity, can be robustly activated by other inputs in N$^{MN\Delta}$6910$^{KO}$ mice. Our results suggest a selective requirement for cadherin function for phrenic MN activation by descending excitatory inputs.

## Cadherin 6/9/10 expression defines the core motor respiratory circuit

While cadherins likely contribute to phrenic MN output by establishing MN morphology, they can also act as recognition molecules between presynaptic and postsynaptic neurons to promote connectivity independently of cell body position or dendritic orientation (*Basu et al., 2017*; *Basu et al., 2015*). To test whether cadherins are also required in presynaptic partners of PMC neurons to generate robust respiratory output, we performed in situ hybridization to define cadherin expression in the rVRG, the major monosynaptic input neurons to phrenic MNs (*Wu et al., 2017*), at e15.5, when rVRG

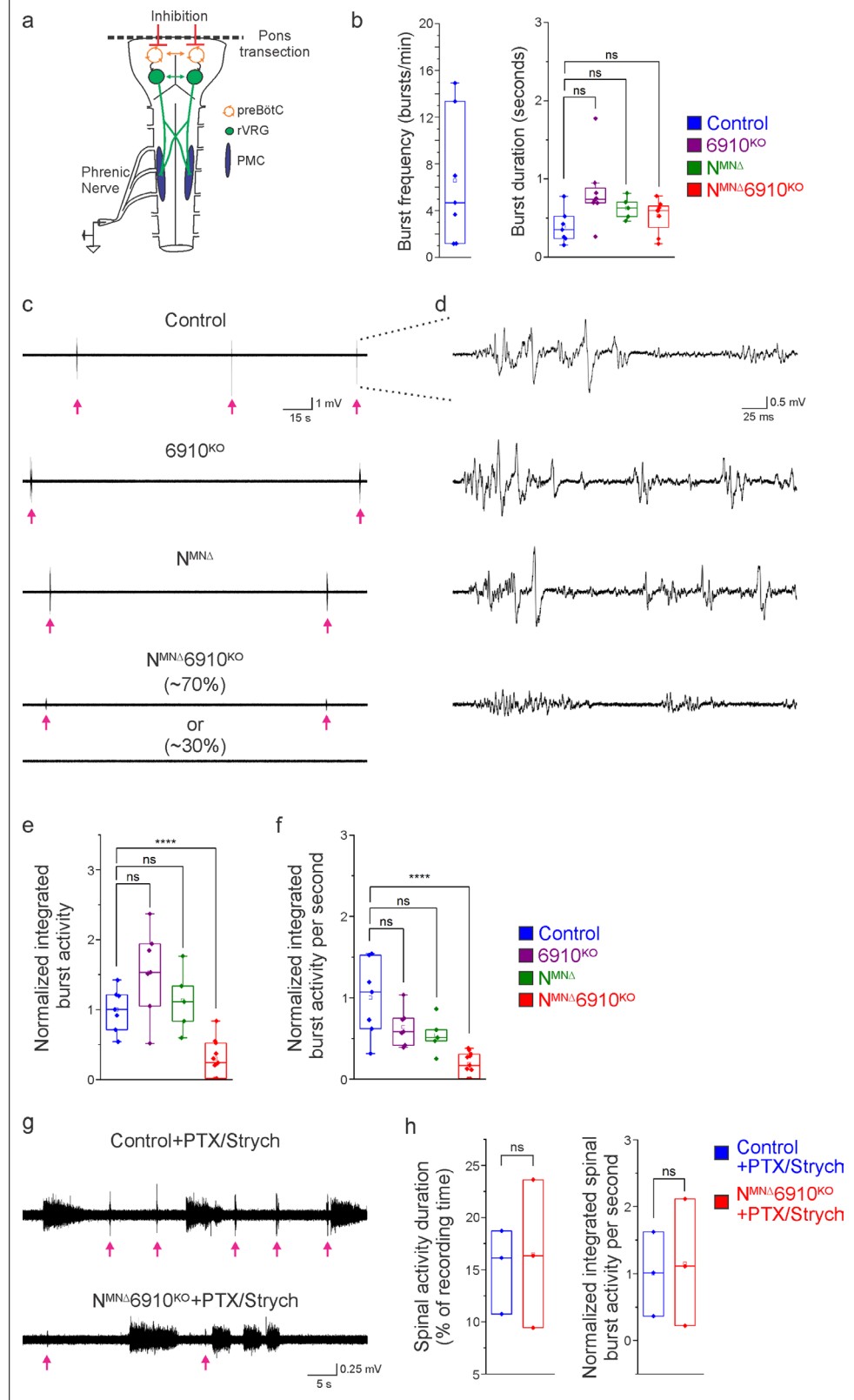

**Figure 4.** Coordinated type I and II cadherin signaling drives phrenic motor neuron (MN) activation. (**a**) Schematic of brainstem–spinal cord preparation, which displays fictive inspiration after removal of the pons. Suction electrode recordings were taken from the phrenic nerve in the thoracic cavity at e18.5/P0. (**b**) Burst frequency is highly variable in control mice at e18.5/P0, as the respiratory rhythm has not yet stabilized, precluding meaningful

*Figure 4 continued on next page*

*Figure 4 continued*

analysis of respiratory frequency amongst different cadherin mutants. Burst duration is similar in all mice at e18.5/P0. (**c**) Three minute recordings from the phrenic nerve in control, 6910$^{KO}$, N$^{MN\Delta}$, and N$^{MN\Delta}$6910$^{KO}$ mice. While 70% of N$^{MN\Delta}$6910$^{KO}$ mice display respiratory bursts, 30% show no bursts throughout the recording period. Respiratory bursts are indicated with magenta arrows. (**d**) Enlargement of single respiratory bursts reveals a reduction in burst amplitude and overall activity in N$^{MN\Delta}$6910$^{KO}$ mice. Partial (initial 350 ms) bursts are shown. (**e, f**) N$^{MN\Delta}$6910$^{KO}$ mice exhibit nearly 70% reduction in normalized integrated burst activity and normalized integrated burst activity over time; see Materials and methods for more information about quantification ($n$ = 7 control, $n$ = 7 6910$^{KO}$, $n$ = 5 N$^{MN\Delta}$, $n$ = 11 N$^{MN\Delta}$6910$^{KO}$ mice). (**g**) Recordings from control and N$^{MN\Delta}$6910$^{KO}$ mice after disinhibition of the brainstem–spinal cord preparation with picrotoxin and strychnine at e18.5/P0. Disinhibition reveals the existence of latent spinal network activity, which is distinct from respiratory bursts (indicated with magenta arrows) and exhibits longer duration. (**h**) Control and N$^{MN\Delta}$6910$^{KO}$ mice display similar spinal network activity duration and normalized integrated activity over time, suggesting that phrenic MNs are normally integrated into propriospinal circuits and, despite the reduction in respiratory activity, can be robustly activated by other inputs in N$^{MN\Delta}$6910$^{KO}$ mice. ****p < 0.0001, unpaired, two-tailed Student's *t*-test.

The online version of this article includes the following source data for figure 4:

**Source data 1.**

**Source data 2.**

---

to PMC connectivity is being established. We found that rVRG neurons express a complementary cadherin code to phrenic MNs, showing specific and robust expression of *Cdh6*, *Cdh9*, and *Cdh10*, while *N-cadherin* is broadly expressed in the brainstem, mirroring the expression pattern seen in the spinal cord (*Figure 5a–g*).

While the rVRG can be anatomically defined, there are currently no known molecular markers for this population. To confirm that the *Cdh9* expression we observe in the brainstem corresponds to the rVRG, we performed rabies retrograde viral circuit tracing. We generated C*dh9*$^{iCre}$ mice using CRISPR-Cas9 technology. Briefly, a *P2A-iCre* sequence (1222 bp) was inserted in place of the stop codon in *Cdh9* exon 12 (see Materials and methods). We confirmed that the *Cdh9*$^{iCre}$ driver induces recombination in phrenic MNs as early as e12.5 and in the brainstem, at the putative location of the rVRG, at e15.5 (*Figure 5—figure supplement 1a, b*). We crossed *Cdh9*$^{iCre}$ mice to Ai3 reporter mice and RphiGT mice, which express G protein after Cre-mediated recombination (*Takatoh et al., 2013*). This allowed us to both fluorescently label all Cdh9-expressing populations and perform rabies virus-mediated retrograde tracing. We injected Rabies$\Delta$G-mCherry virus in the diaphragm of P4 mice to label phrenic MNs and their presynaptic partners (*Figure 5i*). This approach successfully labeled both phrenic MNs and monosynaptically connected neurons within the rVRG in the brainstem (*Figure 5j–l*). We found that 80% of rVRG neurons ($n$ = 136/195 and 301/335 mCherry + neurons from two mice) are also labeled green by Ai3, indicating that the Cdh6/9/10 + neurons we detected in the brainstem at e15.5 likely correspond to the rVRG (*Figure 5k, l*). Our results show that cadherin 6/9/10 expression defines the core motor respiratory circuit, raising the possibility that cadherins are required in multiple respiratory neuronal populations.

## Cadherin signaling is required in Dbx1-derived neurons for robust respiratory output

In order to assess the role of cadherins in brainstem respiratory neuron function, we eliminated cadherin signaling by inactivating β- and γ-catenin using a *Dbx1*$^{Cre}$ promoter (*Ctnnb1*$^{flox/flox}$;*Jup*$^{flox/flox}$;*Dbx1*$^{Cre}$, referred to as βγ-cat$^{Dbx1\Delta}$ mice). β- and γ-catenin are obligate intracellular factors required for cadherin-mediated cell adhesive function (*Figure 6a*). Driving Cre expression using the *Dbx1* promoter eliminates all cadherin signaling in the V$_0$ progenitor domain, which gives rise to multiple brainstem respiratory populations, including the premotor rVRG and the rhythmogenic preBötC. We confirmed that Dbx1-mediated recombination does not target phrenic MNs (*Figure 6—figure supplement 1*). Dbx1-mediated gene ablation has been utilized to demonstrate the mechanisms of left/right coupling in the rVRG (*Wu et al., 2017*). Inactivating β/γ-catenin in Dbx1-derived neurons circumvents potential redundancy that can arise through the expression of multiple cadherins in the brainstem and allows us to establish a cadherin requirement in Dbx1-derived neurons before dissecting individual cadherin function.

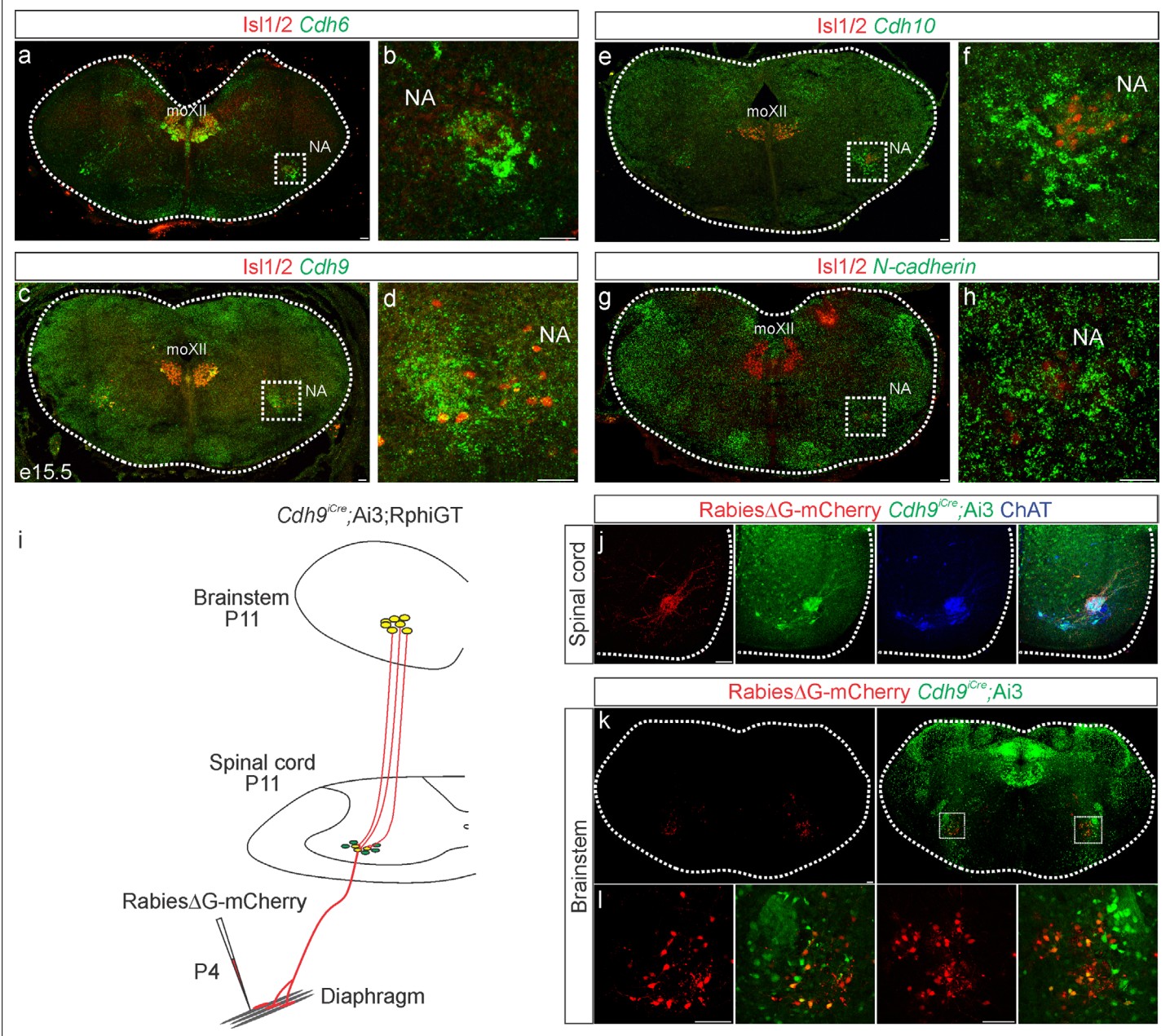

**Figure 5.** Cadherin expression defines the core motor respiratory circuit. Fluorescence in situ hybridization (FISH) showing expression of *Cdh6* (green, **a, b**), *Cdh9* (green, **c, d**), *Cdh10* (green, **e, f**), and *N-cadherin* (green, **g, h**) in the brainstem, at the putative location of the rostral Ventral Respiratory Group (rVRG), the main source of synaptic input to phrenic motor neurons (MNs). The location of the rVRG is inferred based on its relationship to motor nuclei at the same rostrocaudal level of the brainstem, the nucleus ambiguous (NA) and the hypoglossal motor nucleus (moXII), which are labeled by the MN-specific transcription factor (TF) Isl1/2 (red, **a–h**). b, d, f, and h are the enlarged regions outlined in the boxes in a, c, e, and g, respectively. Scale bar = 50 µm. (**i**) Strategy for tracing respiratory motor circuits in neonatal (P4) mice. RabiesΔG-mCherry is injected into the diaphragm of *Cdh9^{iCre}*;Ai3;RphiGT mice. Ai3 labels *Cdh9^{iCre}* expressing cells in green and RphiGT allows for Cre-dependent G protein expression and transsynaptic labeling. (**j**) *Cdh9^{iCre}*-induced recombination in phrenic MNs, demonstrated by Cre-dependent YFP expression (green). All MNs are labeled by Choline Acetyltransferase (ChAT) expression (blue). RabiesΔG-mCherry injection into the diaphragm exclusively infects phrenic MNs (red). Scale bar = 100 µm. (**k, l**) RabiesΔG-mCherry injection into the diaphragm of *Cdh9^{iCre}*;Ai3;RphiGT mice results in transsynaptic labeling of rVRG neurons (red). 80% of rVRG neurons are cdh9+ (green, *n* = 136/195 and 301/335 mCherry + neurons from two mice), demonstrating that a complementary cadherin code defines the core respiratory motor circuit. Scale bar = 100 µm.

The online version of this article includes the following figure supplement(s) for figure 5:

**Figure supplement 1.** Validation of *Cdh9^{iCre}* mice.

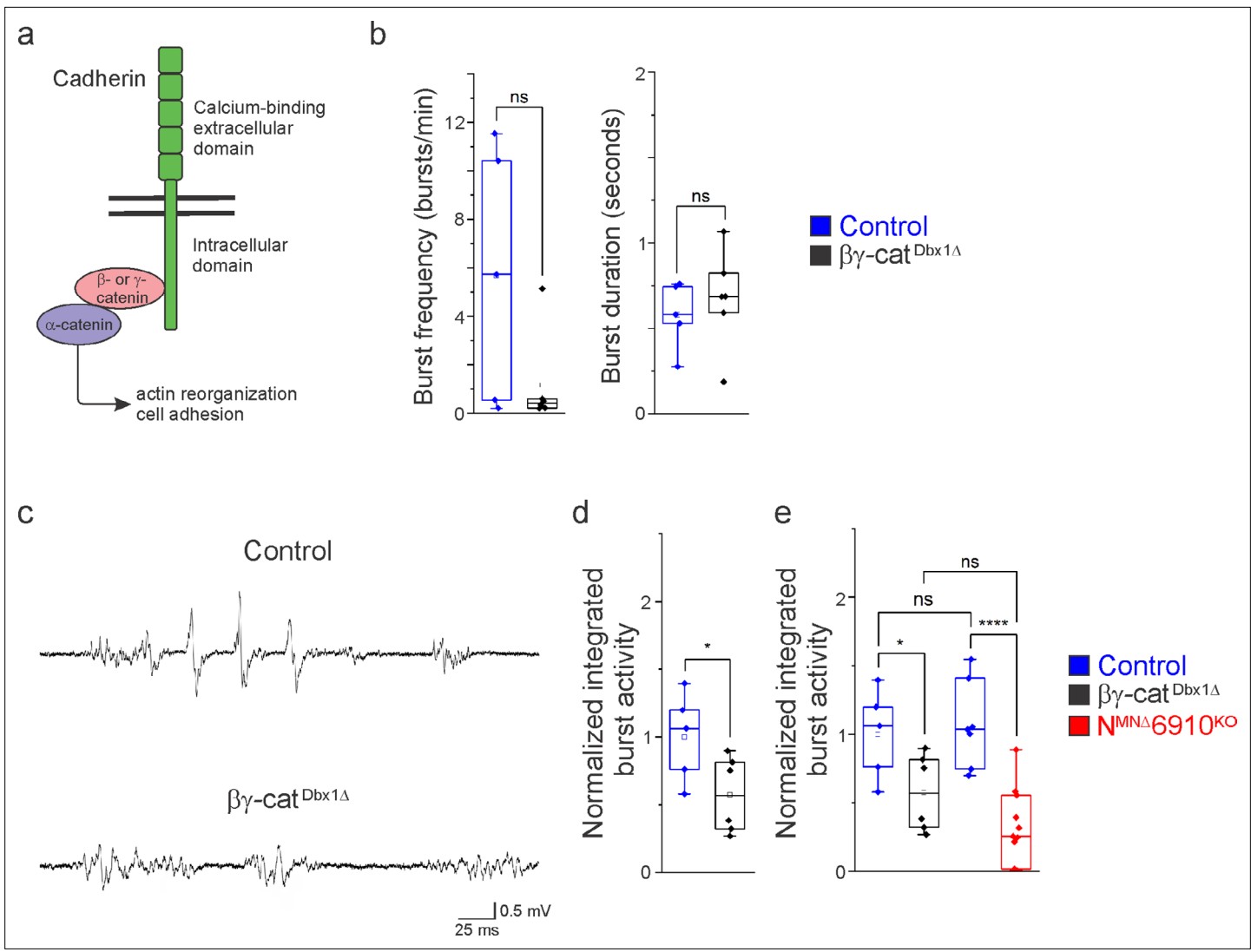

**Figure 6.** Cadherin signaling is required in Dbx1-derived neurons for robust respiratory output. (**a**) B- and γ-catenin are obligate intracellular factors required for cadherin-mediated cell adhesive function. We utilized inactivation of β- and γ-catenin in Dbx1-derived interneurons (*Ctnnb1*$^{flox/flox}$;*Jup*$^{flox/flox}$;*Dbx1*$^{Cre}$, referred to as βγ-cat$^{Dbx1Δ}$ mice) as a strategy to define the function of cadherins in premotor respiratory populations. (**b**) Bγ-cat$^{Dbx1Δ}$ mice do not show changes in burst frequency and duration. (**c**) Suction electrode recordings from the phrenic nerve at e18.5/P0 in control and βγ-cat$^{Dbx1Δ}$ mice. Enlargement of a single respiratory burst reveals a reduction in burst amplitude and overall activity in βγ-cat$^{Dbx1Δ}$ mice. (**d**) βγ-cat$^{Dbx1Δ}$ mice exhibit a 50% reduction in burst activity; see Materials and methods for more information about quantification. (**e**) N$^{MNΔ}$6910$^{KO}$ mice and βγ-cat$^{Dbx1Δ}$ mice display a similar reduction in burst activity (*n* = 12 control, *n* = 6 βγ-cat$^{Dbx1Δ}$, *n* = 11 N$^{MNΔ}$6910$^{KO}$ mice; data are normalized to the control for βγ-cat$^{Dbx1Δ}$ experiments), suggesting that cadherin function is required in both motor neurons (MNs) and Dbx1-derived interneurons for robust respiratory output. *p < 0.05, ****p < 0.0001, unpaired, two-tailed Student's *t*-test.

The online version of this article includes the following source data and figure supplement(s) for figure 6:

**Source data 1.**

**Figure supplement 1.** Dbx1-mediated recombination does not target phrenic motor neurons (MNs).

We found that βγ-cat$^{Dbx1Δ}$ mice are born alive but die within 24 hr after birth. We next wanted to establish whether this lethality was due to respiratory insufficiency, so we performed phrenic nerve recordings at e18.5/P0. Bγ-cat$^{Dbx1Δ}$ mice do not show significant changes in respiratory burst frequency or duration (*Figure 6b*). Interestingly, we found that βγ-cat$^{Dbx1Δ}$ mice show a reduction in burst activity similar to N$^{MNΔ}$6910$^{KO}$ mice (*Figure 6c*). After rectifying and integrating the traces, we saw a 50% reduction in total burst activity in βγ-cat$^{Dbx1Δ}$ mice as compared to control (*Figure 6d*). When comparing βγ-cat$^{Dbx1Δ}$ mice to N$^{MNΔ}$6910$^{KO}$ mice, no significant difference was seen (*Figure 6e*),

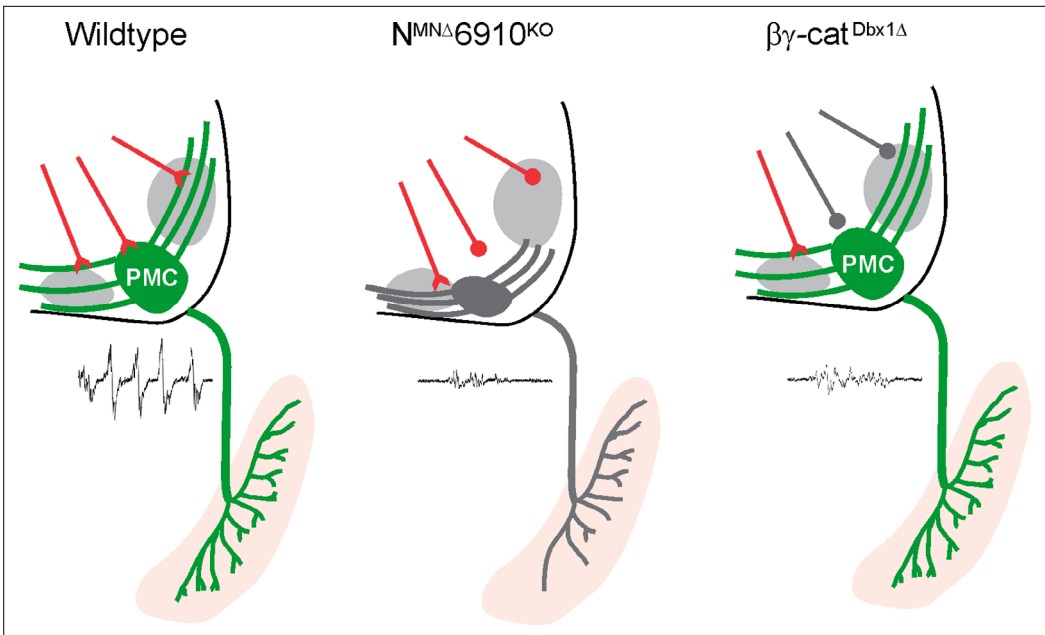

**Figure 7.** Coordinated cadherin functions dictate respiratory motor output. In control animals, phrenic motor neurons (MNs) form a cluster in the cervical, ventral spinal cord and project their dendrites in a stereotypical ventromedial and dorsolateral orientation. This Phrenic Motor Column (PMC) topography is dictated by a combinatorial cadherin code during development. In the absence of cadherins N, 6, 9, and 10, phrenic MN topography is eroded, likely resulting in the selective loss of excitatory inputs and a reduction in respiratory motor output. Inactivation of cadherin signaling in Dbx1-derived neurons, which provide the major input to phrenic MNs, also results in a reduction of phrenic MN output, demonstrating that cadherins are required in both populations for robust activation of phrenic MNs.

suggesting that cadherin function is required both in MNs and Dbx1-derived interneurons for robust respiratory output.

## Discussion

The proper assembly of respiratory motor circuits is imperative for survival, yet the principles that underlie respiratory neuron connectivity are largely unknown. In this study, we find that a combinatorial cadherin code defines the core respiratory motor circuit and that coordinated cadherin function is required in multiple respiratory neuronal populations to drive robust respiratory output and survival (*Figure 7*). MN-specific cadherin deletion alters phrenic MN topography, activity, and breathing behaviors, resulting in death shortly after birth. Similarly, Dbx1-mediated cadherin inactivation also results in diminished phrenic MN output and perinatal death. We discuss these findings in the context of motor circuit assembly and cadherin function.

### Respiratory motor circuit anatomy and assembly

Phrenic MNs are the final output cells of respiratory circuits, providing the only motor innervation to the major inspiratory muscle, the diaphragm. The majority of inputs to phrenic MNs arise from the rVRG in the brainstem (*Wu et al., 2017*), which receives critical inputs from the preBötC, the inspiratory central pattern generator. The circuit connecting the preBötC to the rVRG to the PMC is thus imperative for conveying robust inspiratory drive onto phrenic MNs and initiating diaphragm contractions. Without these synapses, breathing cannot occur.

The excitatory interneurons in both the preBötC and the rVRG arise from the Dbx1[+], p0 progenitor domain (*Bouvier et al., 2010*; *Gray et al., 2010*; *Wu et al., 2017*), while phrenic MNs are derived from the Olig2[+] progenitor MN domain. The role of Dbx1-derived neurons in breathing has been well established (*Del Negro et al., 2018*). For example, Dbx1-specific laser ablation of the preBötC resulted in an impairment in respiratory rhythm, with the respiratory rhythm ceasing completely when

enough cells were ablated. Interestingly, this study also found that the magnitude of motor output was reduced, suggesting some of the cells ablated were functioning in a premotor capacity rather than as a central pattern generator (*Wang et al., 2014*). Further studies identified Dbx1[+] premotor populations that, when ablated, reduced respiratory motor output without affecting frequency (*Revill et al., 2015*). Here, we show that ablation of cadherin signaling in Dbx1[+] neurons also results in a reduction in respiratory output without changes in frequency. As both the preBötC and the rVRG are derived from the Dbx1[+] domain, it is possible that Dbx1-mediated cadherin inactivation alters the function of both respiratory populations. Our finding of a reduced motor output in the setting of normal frequency suggests that the rhythmogenic properties of the preBötC are intact, while the transmission of excitatory drive to phrenic MNs is impaired.

Here, we also show for the first time, that a phrenic MN manipulation can result in a severe reduction or ablation of respiratory motor output. What is the exact mechanism that underlies the reduction in activity seen in N$^{MN\Delta}$6910$^{KO}$ mice? While reduction of phrenic MN output could reflect changes in the intrinsic properties or excitability of phrenic MNs, we think this is unlikely given the selective loss of respiratory activity. In addition to excitatory rVRG inputs, phrenic MNs also receive inputs from propriospinal networks (*Cregg et al., 2017*). This network activity is latent and can only be revealed after disinhibition with GABA$_A$ and glycine receptor antagonists. Interestingly, we observed similar levels of propriospinal network activity after disinhibition in control and N$^{MN\Delta}$6910$^{KO}$ mice. This finding indicates that phrenic MNs are normally integrated into propriospinal circuits and, despite the reduction in respiratory activity, can be robustly activated by other inputs in N$^{MN\Delta}$6910$^{KO}$ mice. Therefore, we favor the hypothesis that reduced phrenic bursting activity reflects a selective loss of excitatory rVRG to PMC connections.

## Cadherins, topography, and circuit formation

Cadherins have well-established roles in early development, such as their function in the migration of developing neurons. For example, cadherins regulate the migration of cortical neurons to their correct layers (*Martinez-Garay, 2020*) and the positioning of MNs in the hindbrain and ocular system (*Astick et al., 2014*; *Knüfer et al., 2020*; *Montague et al., 2017*). In spinal motor circuits, cadherins have largely been examined in terms of their role in positioning limb-innervating MN cell bodies in the lumbar spinal cord (*Bello et al., 2012*; *Demireva et al., 2011*; *Dewitz et al., 2019*; *Dewitz et al., 2018*; *Price et al., 2002*). Here, we show that cadherins also dictate the cell body positioning of phrenic MNs. However, unlike in lumbar MNs, where both type I and II cadherins are required for the organization of MN cell bodies in certain motor pools, we find that phrenic MN positioning is exclusively dependent on N-cadherin, as positional defects in N$^{MN\Delta}$ mice and N$^{MN\Delta}$6910$^{KO}$ mice were similar. Perhaps the PMC, with fewer motor pools comprising it as compared to limb-innervating MNs, requires less complex expression of adhesive molecules to segregate and cluster appropriately.

Surprisingly, we find that the stereotypical topographic position of phrenic MNs is not required for their targeting and robust activation by their descending inputs in the rVRG, which produce periodic respiratory bursts. Despite exhibiting a substantial ventral shift of phrenic MNs, N$^{MN\Delta}$ mice displayed normal phrenic MN bursting activity. The fact that connectivity does not rely on cell body topography seems to be a principle that is evolutionarily conserved in the respiratory system. For example, respiratory activities of zebrafish branchiomotor neurons are resistant to shifts in their rostrocaudal positioning (*McArthur and Fetcho, 2017*). This contrasts with observations on proprioceptive inputs onto limb-innervating MNs where topographic position has been proposed to play a dominant role (*Sürmeli et al., 2011*), suggesting that different strategies may be utilized to construct distinct motor circuits. Independent manipulation of cell body position without affecting transcriptional programs or the cell surface proteome of MNs will ultimately be required to parse out the relative contributions of cell body topography to motor circuit connectivity.

Both type I and II cadherins have been implicated in dendrite morphogenesis and arborization (*Hirano and Takeichi, 2012*). N-cadherin enhances dendritic extension and branching in cultured hippocampal neurons (*Esch et al., 2000*; *Yu and Malenka, 2003*) and treatment of hippocampal slices with a peptide that blocks N-cadherin leads to dendritic retraction in CA3 neurons (*Bekirov et al., 2008*). N-cadherin also promotes dendritic arborization in retinal ganglia (*Riehl et al., 1996*) and horizontal cells (*Tanabe et al., 2006*). In contrast, N-cadherin appears to restrict dendritic growth and targeting in amacrine cells (*Masai et al., 2003*) and olfactory projection neurons (*Zhu and Luo, 2004*),

arguing for cell-type and environment-dependent functions of N-cadherin on dendritic arborization. In the retina, type II cadherins are required for the tight fasciculation of dendrites from direction-selective ganglion cells with starburst amacrine cells (*Duan et al., 2018*). Here, we find no changes in phrenic dendrites in 6910$^{KO}$ mice, and only mild dendritic defects in N$^{MN\Delta}$ mice, characterized by a ventral shift of the most dorsolateral dendrites. It is possible that the shift in cell body position in N$^{MN\Delta}$ mice directly results in dendritic mislocalization. Therefore, it would be interesting to sparsely label individual phrenic MNs in the future to compare the dendritic arbors of cells located in the correct or shifted position. In N$^{MN\Delta}$6910$^{KO}$ mice, phrenic dendrites appear more ventralized and defasciculated than in N$^{MN\Delta}$ mice, suggesting coordinated functions for N-cadherin and type II cadherins in dendritic orientation and fasciculation of phrenic MNs. Our data reveal a novel complex interplay between type I and II cadherins for establishing phrenic MN morphology in vivo.

## Cadherin interactions and molecular recognition

Cadherins have long been hypothesized to act in matching synaptic partners through molecular recognition, but, due to their role in early developmental processes, this hypothesis remains difficult to test. The role of cadherins as molecular recognition molecules is suggested, however, by experiments where morphology is largely preserved but output is greatly altered. For example, in the hippocampus, Cadherin 9 is strongly and specifically expressed in CA3 pyramidal neurons (*Williams et al., 2011*), while Cadherins 6 and 10 are strongly and specifically expressed in CA1 pyramidal neurons. Cadherin expression is required for high magnitude potentiation, which relies on the interaction of Cadherin 6/10 postsynaptic cis dimers with presynaptic Cadherin 9 in trans (*Basu et al., 2017*). Interestingly, this study shows the importance of heterophilic interactions between cadherins at the level of the synapse in dictating circuit output.

In both the retina and the hippocampus, despite using widely divergent strategies, type II cadherins dictate circuit formation independently, without contributions from type I cadherins such as N-cadherin (*Basu et al., 2017*; *Duan et al., 2014*; *Duan et al., 2018*). In the respiratory motor circuit, however, we find that the coordinated functions of both N-cadherin and type II cadherins are required for circuit formation and function. Interestingly, interactions between type I and II cadherins are thought to be prohibited, so direct interactions seem unlikely (*Patel et al., 2006*). Perhaps instead, N-cadherin provides a basal level of adhesion that is necessary for type II cadherins to modulate adhesive strength between dendrites and synapses.

Our experiments in βγ-cat$^{Dbx1\Delta}$ and N$^{MN\Delta}$6910$^{KO}$ mice show an equivalent reduction in activity after the loss of cadherin signaling in both MNs and Dbx1-derived interneurons, suggesting a role for cadherins in synaptic matching between these two populations. However, we cannot exclude the possibility that loss of cadherin signaling may alter the migration, morphology or axon guidance of Dbx1-derived neurons, leading to changes in respiratory output. Cadherin 9 expression in the brainstem is not restricted to the rVRG, but has also been identified in other respiratory neurons (*Yackle et al., 2017*), indicating cadherins may function broadly to establish synaptic specificity throughout the respiratory circuit. Future experiments utilizing restricted genetic labeling of specific respiratory neurons will further define cadherin function in distinct populations.

# Materials and methods
## Mouse genetics

The *Cdh2$^{flox/flox}$* (*Kostetskii et al., 2005*), *Ctnnb1$^{flox/flox}$* (*Brault et al., 2001*), *Jup$^{flox/flox}$* (*Demireva et al., 2011*), *Cdh6−/−;Cdh9−/−;Cdh10−/−* (*Duan et al., 2018*), *Olig2$^{Cre}$* (*Dessaud et al., 2007*), *Chat$^{Cre}$* (*Lowell et al., 2006*), *ROSA26Sor$^{tm9(CAG-tdTomato)}$* (Ai9, JAX# 007909) and *ROSA26$^{CAG-LSL-EYFP-WPRE}$* (Ai3, JAX# 007903) (*Madisen et al., 2010*), *ROSA26$^{tm1(CAG-RABVgp4,-TVA)Arenk}$* (RphiGT, JAX# 024708, *Takatoh et al., 2013*), and *Dbx1$^{Cre}$* (*Bielle et al., 2005*) lines were generated as previously described and maintained on a mixed background. *Cdh9$^{iCre}$* mice were generated using CRISPR-Cas9 technology. Briefly, a P2A-iCre sequence (1222 bp) was inserted in place of the stop codon in mCdh9 exon 12. sgRNAs targeting within 20 bp of the desired integration site were designed with at least 3 bp of mismatch between the target site and any other site in the genome. Targeted integration was confirmed in vitro prior to moving forward with embryo injections. C57BL/6J fertilized zygotes were micro-injected in the pronucleus with a mixture of Cas9 protein at 30–60 ng/μl, and single

guide RNA at 10–20 ng/µl each, and a ssDNA at 5–10 ng/µl (a ribonucleoprotein complex). The injected zygotes, after culture in M16 or alternatively Advanced-KSOM media, were transferred into the oviducts of pseudo-pregnant CD-1 females. Founder mice were genotyped by targeted next generation sequencing followed by analysis using CRIS.py. Mouse colony maintenance and handling was performed in compliance with protocols approved by the Institutional Animal Care Use Committee of Case Western Reserve University (assurance number: A-3145-01, protocol number: 2015-0180). Mice were housed in a 12-hr light/dark cycle in cages containing no more than five animals at a time.

## Immunohistochemistry and in situ hybridization

In situ hybridization, fluorescence in situ hybridization, and immunohistochemistry were performed as previously described (*Philippidou et al., 2012*; *Vagnozzi et al., 2020*), on tissue fixed for 2 hr in 4% paraformaldehyde (PFA) and cryosectioned at 16 µm. In situ probes were generated from e12.5 cervical spinal cord cDNA libraries using PCR primers with a T7 RNA polymerase promoter sequence at the 5′ end of the reverse primer. All probes generated were 750–1000 bp in length. Transcripts are either indicated by a black-colored precipitate resulting from alkaline-phosphatase colorimetric reactions (*Figure 1—figure supplement 1*) or by fluorescence signal (*Figure 5*). Wholemounts of diaphragm muscles were stained as described (*Philippidou et al., 2012*). The following antibodies were used: goat anti-Scip (1:5000; Santa Cruz Biotechnology, RRID:AB_2268536), mouse anti-Islet1/2 (1:1000, DSHB, RRID:AB_2314683) (*Tsuchida et al., 1994*), rabbit anti-neurofilament (1:1000; Synaptic Systems, RRID:AB_887743), rabbit anti-synaptophysin (1:250, Thermo Fisher, RRID:AB_10983675), α-bungarotoxin Alexa Fluor 555 conjugate (1:1000; Invitrogen, RRID:AB_2617152), and goat anti-ChAT (1:200, Millipore, RRID:AB_2079751). Images were obtained with a Zeiss LSM 800 confocal microscope and analyzed with Zen Blue, ImageJ (Fiji), and Imaris (Bitplane). Phrenic MN number was quantified using the Imaris 'spots' function to detect cell bodies that coexpressed high levels of Scip and Isl1/2 in a region of interest limited to the left and right sides of the ventral spinal cord. Diaphragm innervation was quantified using the simple neurite tracer plugin in ImageJ.

## DiI tracing

For labeling of phrenic MNs, crystals of carbocyanine dye, DiI (Invitrogen, #D3911) were pressed onto the phrenic nerves of eviscerated embryos at e18.5, and the embryos were incubated in 4% PFA at 37°C in the dark for 4–5 weeks. Spinal cords were then dissected, embedded in 4% low melting point agarose (Invitrogen) and sectioned using a Leica VT1000S vibratome at 100–150 µm.

## Positional analysis

MN positional analysis was performed as previously described (*Dewitz et al., 2019*; *Dewitz et al., 2018*). MN positions were acquired using the 'spots' function of the imaging software Imaris (Bitplane) to assign *x* and *y* coordinates. Coordinates were expressed relative to the midpoint of the spinal cord midline, defined as position $x = 0$, $y = 0$. To account for experimental variations in spinal cord size, orientation, and shape, sections were normalized to a standardized spinal cord whose dimensions were empirically calculated at e13.5 (midline to the lateral edge = 390 µm). Contour plots and histograms include all the combined somas from $n = 3$ control, $n = 3$ 6910$^{KO}$, $n = 3$ N$^{MNΔ}$, and $n = 4$ N$^{MNΔ}$6910$^{KO}$ mice. This includes $n = 1442$ control, $n = 1422$ 6910$^{KO}$, $n = 1052$ N$^{MNΔ}$, and $n = 1220$ N$^{MNΔ}$6910$^{KO}$ somas. Somas were counted from both L and R sides of the spinal cord from every other 16 µm section that contained the PMC (20–30 sections per embryo), and thus represent half the total number of somas on L and R sides.

## Dendritic orientation analysis

For the analysis of dendritic orientation, we superimposed a radial grid divided into eighths (45 degrees per octant) centered over phrenic MN cell bodies spanning the entire length of the dendrites. We drew a circle around the cell bodies and deleted the fluorescence associated with them. Fiji (ImageJ) was used to calculate the fluorescent intensity (IntDen) in each octant which was divided by the sum of the total fluorescent intensity to calculate the percentage of dendritic intensity in each area.

## Rabies virus production and retrograde tracing

Rabies virus was produced in B7GG cells stably expressing G protein for complementation and HEK293t cells to determine viral titers as described previously (*Wickersham et al., 2007*). RabiesΔG-mCherry virus (titer of around $1^{10}$ TU/ml) was mixed at a 2:1 ratio with silk fibroin (Sigma #5154) to make rabies injection solution (*Jackman et al., 2018*). Neonatal mice (P4) were anesthetized on ice before injection. 1.5 μl of rabies injection solution was injected into one side of the diaphragm of P4 *Cdh9iCre*;RphiGT;Ai3 mice with a glass electrode using a nano-injector (Drummond). *Cdh9iCre*;RphiGT;Ai3 mice express Rabies G-protein in a *Cdh9iCre*-dependent manner, allowing for transsynaptic transport of RabiesΔG-mCherry virus from Cdh9-expressing neurons. 7 days postinjection, P11 mice were sacrificed after being anesthetized by intraperitoneal injection of a ketamine/xylazine cocktail solution. The mCherry fluorescent signal in the diaphragm and spinal cord (from cervical to thoracic levels) was checked to ensure specific labeling of one side of the diaphragm/phrenic MNs. Brainstem and spinal cord tissue were fixed in 4% PFA overnight, followed by phosphate-buffered saline washes. The tissue was then embedded in 4% low melting point agarose (Invitrogen) and sectioned using a Leica VT1000S vibratome at 100 μm.

## Electrophysiology

Electrophysiology was performed as previously described (*Vagnozzi et al., 2020*). Mice were cryoanesthetized and rapid dissection was carried out in 22–26°C oxygenated Ringer's solution. The solution was composed of 128 mM NaCl, 4 mM KCl, 21 mM NaHCO$_3$, 0.5 mM NaH$_2$PO$_4$, 2 mM CaCl$_2$, 1 mM MgCl$_2$, and 30 mM D-glucose and was equilibrated by bubbling in 95% O$_2$/5% CO$_2$. The hindbrain and spinal cord were exposed by ventral laminectomy, and phrenic nerves exposed and dissected free of connective tissue. A transection at the pontomedullary boundary rostral to the anterior inferior cerebellar artery was used to initiate fictive inspiration. Electrophysiology was performed under continuous perfusion of oxygenated Ringer's solution from rostral to caudal. Suction electrodes were attached to phrenic nerves just proximal to their arrival at the diaphragm. We bath applied the following drugs: picrotoxin (PTX) (GABA$_A$ receptor antagonist, 10 μM, Tocris Bioscience, #1128) and strychnine hydrochloride (Strych) (glycine receptor antagonist, 0.3 μM, Sigma, #S8753) dissolved in Ringer's solution. The signal was band-pass filtered from 10 Hz to 3 kHz using AM-Systems amplifiers (Model 3000), amplified 5000-fold, and sampled at a rate of 50 kHz with a Digidata 1440A (Molecular Devices). Data were recorded using AxoScope software (Molecular Devices) and analyzed in Spike2 (Cambridge Electronic Design). Burst duration and burst activity were computed from 7 bursts per mouse, except for 4 out of 11 N$^{MNΔ}$6910$^{KO}$ mice, which only had 1–3 bursts total. For data analysis, either every single burst was analyzed, or for those traces of higher frequency, bursts were selected randomly, spaced throughout the trace. Bursts were defined as activity above baseline that persists for at least 50 ms. Some bursts contain pauses in activity in the middle; activity that was spaced less than 1 s apart was defined as a single burst. Burst frequency was determined from 10 or more minutes of recording time per mouse. Burst activity was computed by rectifying and integrating the traces with an integration time equal to 2 s, long enough to encompass the entire burst. The maximum amplitude of the rectified and integrated signal was then measured and reported as the total burst activity or divided by burst duration and reported as normalized integrated activity over time. For quantitation of propriospinal activity after disinhibition, we used 6–11 bursts per mouse from 3 control and 3 N$^{MNΔ}$6910$^{KO}$ mice to calculate the total spinal burst duration (shown as % of recording time) and the normalized integrated spinal activity over time, as these bursts show variable duration.

## Plethysmography

Conscious, unrestrained P0–P2 mice were placed in a whole body plethysmograph (emka) attached to a differential pressure transducer (emka). We modified 10-ml syringes to use as chambers, as smaller chambers increase signal detection in younger mice. Experiments were done in room air (79% nitrogen, 21% oxygen). Mice were placed in the chamber for 30 s at a time, for a total of three to five times, and breathing parameters were recorded. Mice were directly observed to identify resting breaths. Moments of quiet breathing with no movement (which can create noise) were notated and selected for analysis. At least 10 resting breaths were analyzed from every mouse. Data are presented as fold control, where the control is the average of two littermates in normal air. Control mice for N$^{MNΔ}$6910$^{KO}$ experiments were all 6910$^{KO}$, as 6910$^{KO}$ mice survive to adulthood and show normal minute ventilation

(volume of air inhaled per minute). The traces shown represent air flow. The *y*-axis represents air flow (ml/s), while the *x*-axis is in seconds. Inspiration is represented by downward deflection/negative flow numbers, while expiration is represented by upward deflection/positive flow numbers. 0 represents zero flow, such as at the transition between inspiration and expiration or vice versa. Each breath is defined from where it crosses the *x*-axis (*y* = 0). Tidal volume is represented by the area under the curve of the inspiratory phase (see *Figure 1—figure supplement 2a*), and it is automatically calculated by emka software.

## Experimental design and statistical analysis

For all experiments a minimum of three embryos per genotype, both male and female, were used for all reported results unless otherwise stated. Data are presented as box and whisker plots with each dot representing data from one mouse unless otherwise stated. Small open squares in box and whisker plots represent the mean. p values were calculated using unpaired, two-tailed Student's *t*-test. $p < 0.05$ was considered to be statistically significant, where $*p < 0.05$, $**p < 0.01$, $***p < 0.001$, and $****p < 0.0001$.

## Acknowledgements

We thank Heather Broihier, Evan Deneris, Ashleigh Schaffer, Helen Miranda, Jerry Silver, and members of the Philippidou lab for helpful discussions and comments on the manuscript. We thank Ben Deverman for suggesting the silk/rabies virus injection mix and Sebnem Tuncdemir for help with rabies virus production. This work was funded by NIH R01NS114510 to PP, R01EY030138 to XD, institutional funds from St. Jude Children's Research Hospital to LAS, F30HD096788 to ANV, T32GM007250 to ANV/CWRU MSTP, F31NS124240 to MTM, F31NS120699 and T32GM008056 to RKC. PP is the Weidenthal Family Designated Professor in Career Development.

## Additional information

### Funding

| Funder | Grant reference number | Author |
|---|---|---|
| National Institute of Neurological Disorders and Stroke | R01NS114510 | Polyxeni Philippidou |
| National Eye Institute | R01EY030138 | Xin Duan |
| Eunice Kennedy Shriver National Institute of Child Health and Human Development | F30HD096788 | Alicia N Vagnozzi |
| National Institute of Neurological Disorders and Stroke | F31NS124240 | Matthew T Moore |
| National Institute of Neurological Disorders and Stroke | F31NS120699 | Ritesh KC |
| St. Jude Children's Research Hospital | | Lindsay A Schwarz |
| National Institute of General Medical Sciences | T32GM007250 | Alicia N Vagnozzi |
| National Institute of General Medical Sciences | T32GM008056 | Ritesh KC |

The funders had no role in study design, data collection, and interpretation, or the decision to submit the work for publication.

## Author contributions
Alicia N Vagnozzi, Conceptualization, Investigation, Methodology, Writing - original draft, Writing - review and editing; Matthew T Moore, Minshan Lin, Elyse M Brozost, Ritesh KC, Aambar Agarwal, Investigation; Lindsay A Schwarz, Xin Duan, Resources; Niccolò Zampieri, Resources, Methodology; Lynn T Landmesser, Resources, Investigation, Methodology; Polyxeni Philippidou, Conceptualization, Resources, Supervision, Funding acquisition, Investigation, Methodology, Writing - original draft, Writing - review and editing

## Author ORCIDs
Alicia N Vagnozzi (i) http://orcid.org/0000-0002-6152-8728
Niccolò Zampieri (i) http://orcid.org/0000-0002-2228-9453
Polyxeni Philippidou (i) http://orcid.org/0000-0002-0733-3591

## Ethics
Mouse colony maintenance and handling were performed in compliance with protocols approved by the Institutional Animal Care Use Committee of Case Western Reserve University (assurance number: A-3145-01, protocol number: 2015-0180). Mice were housed in a 12-hr light/dark cycle in cages containing no more than five animals at a time.

## Decision letter and Author response
Decision letter https://doi.org/10.7554/eLife.82116.sa1
Author response https://doi.org/10.7554/eLife.82116.sa2

# Additional files

## Supplementary files
• MDAR checklist

## Data availability
All data generated or analyzed during this study are included in the manuscript and supporting files; source data files have been provided for Figures 1–4 and 6.

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
