## [Editor Report]

The overarching hypothesis is that cadherin adhesion molecules specify the code that enables the premotor brainstem breathing circuits to innervate the phrenic motor neurons that control the primary breathing muscle, the diaphragm. This concept is important for understanding how the breathing control circuit is established and in general, how motor circuitry is developed. This is an extremely thorough investigation of the role of cadherins in generating a functional motor circuit.

---

## [Decision Letter]

**Decision letter after peer review:**

Thank you for submitting your article "Coordinated cadherin functions sculpt respiratory motor circuit connectivity" for consideration by *eLife*. Your article has been reviewed by 3 peer reviewers, and the evaluation has been overseen by a Reviewing Editor and Marianne Bronner as the Senior Editor. The following individual involved in review of your submission has agreed to reveal their identity: Deanna Benson, (Reviewer #2).

All three reviewers appreciated the thorough approach to revealing the functions of cadherins in the development and function of this respiratory circuit.

Only one point would be ideally addressed with additional data. Two of the reviewers raised the concern that the study never directly shows that loss of the cadherins specifically affects the connectivity of the rVRG to the phrenic motor neurons. In the reviews they raised alternative mechanisms that could underlie the observed data. During the discussion phase of the review process, the reviewers talked at length with one another about what experiments might be best to address this point. However they were cognizant that some of the suggested experiments might be too time consuming or difficult to constitute a reasonable request for additional data in this revision. In the end, the reviewers agreed that if the mutant mice were available to perform something like a viral tracing experiment that could quantify connectivity between these two brain regions, then this experiment would significantly strengthen the study. However they also agreed that the best approach to address this concern is up to you to decide.

*Reviewer #1 (Recommendations for the authors):*

The authors must provide more detail in the figure legends and methods for how the measurements were made.

1) Figure 1 part D – what is the trace showing? changes in airflow? pressure? What does 0 represent and what is inspiration vs expiration. The methods do not describe the parameters used to calculate tidal volume, how to define what a breath is, and which breaths were chosen for the analysis.

2) Figure 2 – are the contour plots and histograms from a single animal or multiple? What is the total number of somas counted to create them?

3) Figure 4 – The authors state that the in vitro frequency in figure 4 is inaccurate, but then the in vitro frequency is used to claim the preBötC is not impacted in Dbx1 mutants (conclusion section "respiratory motor circuit anatomy and assembly"). To directly assess this conclusion, the bursting frequency of the in vitro preBötC rhythm should be measured.

4) Figure 5 – the authors say that 70% of the rVRG premotor neurons are Cdh9-cre derived or expressing. This is never quantified. How many cells counted? How many animals?

5) Figure 6 part D – the two box & whisker plots are normalized to different control recordings.

6) Figure S1 – does the black represent gene expression?

---

## [Author Response]

Reviewer #1 (Recommendations for the authors):The authors must provide more detail in the figure legends and methods for how the measurements were made.1) Figure 1 part D – what is the trace showing? changes in airflow? pressure? What does 0 represent and what is inspiration vs expiration. The methods do not describe the parameters used to calculate tidal volume, how to define what a breath is, and which breaths were chosen for the analysis.

The traces represent air flow. The y-axis scale is in mL/sec, while the x-axis is in seconds. Inspiration is represented by downward deflection/negative flow numbers, while expiration is represented by upward deflection/positive flow numbers. 0 represents zero flow, such as at the transition between inspiration and expiration or vice versa. Tidal volume is represented by the area under the curve of the inspiratory phase (see Figure 1—figure supplement 2a). It is automatically calculated by emka specialized software. Note that in the initial version of the manuscript we had presented inspiration as upward deflection-we have now updated our traces to show the conventional breath depiction as described here. Each breath is defined from where it crosses the x-axis (y=0). Breaths to analyze were selected by observing the mouse during recordings. Moments of quiet breathing with no movement (which can create noise) were notated and selected for analysis.

2) Figure 2 – are the contour plots and histograms from a single animal or multiple? What is the total number of somas counted to create them?

Contour plots and histograms include all the combined somas from n=3 control, n=3 6910^KO^, n=3 N^MNΔ^, and n=4 N^MNΔ^6910^KO^ mice. This includes n=1442 control, n=1422 6910^KO^, n=1052 N^MNΔ^, and n=1220 N^MNΔ^6910^KO^ somas. Somas were counted from both L and R sides of the spinal cord from every other 16μm section that contained the PMC (20-30 sections per embryo), and thus represent half the total number of somas on L and R sides.

3) Figure 4 – The authors state that the in vitro frequency in figure 4 is inaccurate, but then the in vitro frequency is used to claim the preBötC is not impacted in Dbx1 mutants (conclusion section "respiratory motor circuit anatomy and assembly"). To directly assess this conclusion, the bursting frequency of the in vitro preBötC rhythm should be measured.

We have now included the quantitation of respiratory frequency data for control and βγ-cat^Dbx1∆^ mice, showing that there are no significant changes in burst frequency in βγ-cat^Dbx1∆^ mice. However, we do agree with the reviewer that the loss of excitatory drive could be due to changes either in the rVRG or the preBötC and we have toned down our conclusions to indicate that the preBötC could be impacted in βγ-cat^Dbx1∆^ mice.

4) Figure 5 – the authors say that 70% of the rVRG premotor neurons are Cdh9-cre derived or expressing. This is never quantified. How many cells counted? How many animals?

In the initial version of the manuscript we showed data from 1 animal, where we counted 195 mCherry+ neurons, out of which 136 were gfp+ (70%). We have now repeated the experiment and quantitation in a second animal, where 301/335 (90%) of mCherry+ traced neurons are Cdh9-Cre derived. We have updated our results and added this information in the figure legend.

5) Figure 6 part D – the two box & whisker plots are normalized to different control recordings.

We have now normalized all data to the same control, the control for βγ-cat^Dbx1∆^ experiments. This has not impacted the data.

6) Figure S1 – does the black represent gene expression?

We have indicated in the methods that the black color in in-situ experiments represents gene expression. We have also included a figure (Figure 1—figure supplement 1a; modified from Vagnozzi et al., 2020) that indicates cadherin expression in e13.5 control mice for comparison.